# Electron Attachment to 5-Fluorouracil: The Role of Hydrogen Fluoride in Dissociation Chemistry

**DOI:** 10.3390/ijms23158325

**Published:** 2022-07-28

**Authors:** Eugene Arthur-Baidoo, Gabriel Schöpfer, Milan Ončák, Lidia Chomicz-Mańka, Janusz Rak, Stephan Denifl

**Affiliations:** 1Institut für Ionenphysik und Angewandte Physik, Leopold-Franzens Universität Innsbruck, Technikerstraße 25, A-6020 Innsbruck, Austria; earthurbaidoo@ucsd.edu (E.A.-B.); gabriel.schoepfer@uibk.ac.at (G.S.); 2Center for Molecular Biosciences Innsbruck, Universität Innsbruck, Technikerstraße 25, A-6020 Innsbruck, Austria; 3Laboratory of Biological Sensitizers, Physical Chemistry Department, Faculty of Chemistry, University of Gdańsk, Wita Stwosza 63, 80-308 Gdańsk, Poland; lidia.chomicz-manka@ug.edu.pl (L.C.-M.); janusz.rak@ug.edu.pl (J.R.)

**Keywords:** electron attachment, 5-fluorouracil, anion, hydrogen fluoride

## Abstract

We investigate dissociative electron attachment to 5-fluorouracil (5-FU) employing a crossed electron-molecular beam experiment and quantum chemical calculations. Upon the formation of the 5-FU^−^ anion, 12 different fragmentation products are observed, the most probable dissociation channel being H loss. The parent anion, 5-FU^−^, is not stable on the experimental timescale (~140 µs), most probably due to the low electron affinity of FU; simple HF loss and F^−^ formation are seen only with a rather weak abundance. The initial dynamics upon electron attachment seems to be governed by hydrogen atom pre-dissociation followed by either its full dissociation or roaming in the vicinity of the molecule, recombining eventually into the HF molecule. When the HF molecule is formed, the released energy might be used for various ring cleavage reactions. Our results show that higher yields of the fluorine anion are most probably prevented through both faster dissociation of an H atom and recombination of F^−^ with a proton to form HF. Resonance calculations indicate that F^−^ is formed upon shape as well as core-excited resonances.

## 1. Introduction

Pyrimidines, either modified or halogenated, have been investigated due to their extraordinary biological activity [1,2,3,4]. In some cases, certain analogs like bromouracil (BrU) are used to replace thymine in DNA and have shown the ability to increase the sensitivity of DNA to radiation damage [1]. Halogenated pyrimidines are compounds that have advanced in their potential use as radiosensitizers in radiotherapy. Such derivatives are formed by replacing the C5-hydrogen with a halogen atom (fluorine, chlorine, bromine, iodine) [5]. These radiosensitizers are known for their incorporation into DNA with further post-irradiation enhancement in cytotoxicity and the role played in DNA repair among tumor cells [6]. The radiosensitization abilities of such compounds may be influenced by electron attachment and dissociation cross sections [4,7]. Their capacity to trap electrons has been associated with the efficient decomposition of their respective anions to reactive intermediates.

It has been established that the interaction of ionizing radiation with a biological medium releases a large number of secondary species, of which electrons are considered the most abundant [8,9]. Electrons with kinetic energies of up to 15 eV are known to play a role in DNA damage upon attachment. The ability of these free electrons to induce strand breaks and other types of damage in plasmid DNA was reported by Sanche and co-workers [9,10,11,12,13]. Low-energy electrons (LEEs) can induce selective fragmentation in molecules through the dissociative electron attachment mechanism, which begins with the initial formation of a transient negative ion (TNI) state and further decomposition of the TNI into a negatively charged fragment and neutral counterpart(s). The reaction pathways may either follow a cleavage of a simple bond or multi-bond cleavages, which may sometimes result in molecular rearrangement.

Experimental and theoretical studies on the action of low-energy electrons have shown that the modification of nucleobases and other DNA subunits is a promising tool in enhancing radiation-induced killing [14,15]. The mechanism of action of these compounds is mainly due to the cleavage of the C5–X bond (where X is the halogen atom and C5 belongs to the parent compound such as uracil), leading to the release of a halide anion X^−^ [16]. For instance, the attachment of LEEs to BrU results in the loss of bromine anion (Br^−^), which leads to the formation of the reactive uracil-5-yl radical by electron-attachment induced dissociation [17]. Subsequently, the abstraction of hydrogen by the uracil-5-yl radical from nearby molecules such as sugar moiety causes a single-strand DNA break [18]. It has also been shown that the uracil-5-yl radical may react with a water molecule to form a hydroxyl radical [15,19].

In the present study, we focus our attention on 5-fluorouracil (C_4_FH_3_N_2_O_2_), hereon referred to as 5-FU or FU. 5-FU is a uracil derivative with the hydrogen atom at the C5-position substituted for a fluorine atom, see Figure 1a. It functions as an antimetabolite drug and is widely used in cancer treatment for colorectal cancer, breast cancer, etc., either as a stand-alone or in combination with other anticancer drugs [20,21]. In a tumor environment, 5-FU acts as an anticancer drug via the inhibition of thymidylate synthase and the incorporation of its metabolites into RNA and DNA [20].

In addition to a study on temporary negative ion states using electron transmission spectroscopy [22] and a study on negative ion formation upon electron transfer collisions [23], DEA to 5-FU has been studied using various experimental setups. In the mass spectrometric study by Abdoul–Carime et al. [5], the authors reported the formation of an anion near the mass of parent anion FU^−^ as well as other anions such as NCO^−^, H_2_C_3_NO^−^, CN^−^ and CN_2_OH^−^ in the electron energy range of 0–18 eV. The ion yield near the parent mass was ambiguously assigned to FU^−^ and/or the dehydrogenated parent anion (FU-H)^−^. The ambiguity of ion yield composed of a few overlapping features in the electron energy region below 2 eV was resolved as the dehydrogenated parent anion in a later study by Abouaf et al. [4]. Furthermore, they reported the formation of an anion due to the loss of HF, which was not reported by Abdoul–Carime et al. [5]. On the other hand, both studies reported the very efficient formation of the halogen anions (Br^−^, Cl^−^, I^−^) from BrU, ClU, and IU, respectively, but none of F^−^ in the case of FU. This result seems to be unexpected since the experimental value of the electron affinity (EA) of the fluorine atom (3.401191 ± 0.000026 eV [24]) is similar to that of Br (3.363583 ± 0.000044 eV [25]) and higher than that of I (3.05900 ± 0.00010 eV [26]). Experimental DEA studies with other fluorinated compounds, such as gemcitabine and fluoroadenine, also did not report the formation of F^−^ [27,28], while for other fluorinated organic ring molecules, its formation was compound specific [29,30,31].

Previously, Wetmore et al. [32] investigated BrU, ClU, and FU concerning their electron affinities, ionization potentials, and dissociation in the gas phase and solution by computational methods, positioning the electron affinity of 5-FU close to zero. It was further reported that the energy required to induce the dissociation of Br^−^ (0.55 eV) from BrU^−^ is less compared to Cl^−^ (0.87 eV) and F^−^ (2.25 eV) from the respective halouracil anions. These results indicate that the release of the halogen anion from 5-FU^−^ is less favorable compared to other halouracils and FU is not susceptible to the electron-induced decomposition with electron energy close to zero eV, which may attach with high cross-sections [33,34]. On the other hand, the calculated threshold for FU indicates that F^−^ formation should be possible above the electron energy of 1.80 eV [32].

In the present study, we report a comprehensive study of DEA to 5-FU in the gas phase. We report the observation of eight fragment anions upon electron attachment to 5-FU that have not been reported in previous studies. We also provide detailed reaction pathways after electron attachment obtained through quantum chemical calculations and rationalize experimental observations. We suggest that roaming of hydrogen (and possibly of F^−^ as well) plays an important role in HF formation, with the released energy enabling further ring opening channels. Finally, we calculate the positions of shape resonances for the F^−^ release, which corresponds well to the experimentally observed features in the anion yield curve.

## 2. Results and Discussion

Upon electron attachment to 5-FU, we observe 12 anionic fragments. The overall results of the detected anions along with resonance maxima and experimental as well as calculated threshold energies are summarized in Table 1. The calculated threshold energies are below the experimental ones with the exception of ions with masses of 129 u and 110 u, as discussed below.

At the CCSD(T)/aug-cc-pVDZ//B3LYP/aug-cc-pVDZ level, the vertical electron affinity of 5-FU to form a valence state is calculated as −0.44 eV, i.e., the anion is predicted to be less stable compared to the neutral molecule, the adiabatic electron affinity is calculated as 0.23 eV. Upon electron attachment to 5-FU, the calculated valence state of the anion possesses the odd electron in a π* orbital of A″ irreducible representation (Figure 1b). When a larger basis set is used at the CCSD(T)/aug-cc-pVDZ(C,N,O,F),TZ(H)+ level of theory (see the Methods section), a dipole-bound state of A′ symmetry is obtained at −0.05 eV with respect to the neutral 5-FU molecule. The dipole moment of the neutral 5-FU molecule is calculated as 4.2 Debye at the B3LYP/aug-cc-pVDZ level used for optimization, enough to support the dipole-bound state.

Figure 2a shows the measured anion efficiency curve for the formation of the anion with mass 129 u attributed to (FU–H)^−^. Herein, we limit the electron energy range only up to 6 eV since no resonance was observed beyond this energy. In order to solve the ambiguity in the previous reports, we compare the current results to the anion efficiency curve reported as FU^−^/(FU–H)^−^ by Abdoul–Carime et al. [5] and (FU–H)^−^ by Abouaf et al. [4]. In both studies, the anion efficiency curve showed a sharp peak around 0.6 eV and other resonances below 5 eV. A broad resonance was also reported at around 6 eV, followed by ion-pair formation beyond 10 eV [5]. A comparison of the shape of Figure 2a with both reports shows characteristic resonance positions except for the higher ion beam intensity recorded in the present study. The high resolution of our experimental setup made it possible to resolve the resonances such that the intermediate resonances near 1 eV and 3.7 eV are distinguished, unlike in the previously reported data [5]. Even higher energy resolution used in Ref. [4] allowed to resolve the flat-top structure at about 1.5 eV in two features. At 130 u, we observe ion yield with the same features as that of (FU–H)^−^ and with intensities matching the expected isotope ratio. Therefore, we infer from the present results that the anion yield at the parent mass can only be associated with the dehydrogenated parent (FU–H)^−^ anion. The fact that no parent anion is observed in the experiment suggests that, under isolated conditions in the gas phase, the electron detaches on the experimental time scale unless other reactions take place beforehand. It is also in agreement with the calculated negative vertical electron affinity to form a valence state. In a previous study on photoelectron spectroscopy, it was possible to generate the FU^−^ anion by injecting low-energy electrons into an expansion of FU and argon used as seeding gas [35]. The timescale for detection of anions was ~10 µs. In our experiment, the anions require a lifetime of about ~140 µs to reach the channeltron detector. Thus, the lifetime of the FU^−^ anion towards autodetachment seems too low for direct detection in our experiment.

The molecular mechanism of the (FU–H)^−^ formation is analyzed in Figure 3. First, FU^−^ relaxes into the closest minimum with the energy of −0.23 eV with respect to the neutral FU molecule. The direct pathway of H dissociation from the molecule requires 0.64 eV (0.40 eV at the B3LYP level). The seemingly experimental onset of H loss is ~0.0 eV; the calculated value is thus about 0.6 eV too high. However, the signal is very weak up to about 0.4 eV, suggesting that till this electron energy, only the fraction of ions with considerable thermal energy may release the hydrogen (average thermal energy at the experimental temperature is calculated as 0.33 eV). Alternatively, this signal may also be assigned to an experimental artifact. As noted above, the lifetime of the FU^−^ anions is comparably long, i.e., FU^−^ anions formed will be extracted towards the quadrupole mass spectrometer. As a consequence, electrons that are spontaneously emitted from the metastable parent anion in the region between the interaction region and the entrance of the quadrupole will be accelerated as well, and they may induce DEA on neutral molecules flying in parallel to the anion beam. At higher electron energies (~0.5–3.0 eV), hydrogen loss becomes the most important DEA channel, in agreement with the calculated low dissociation energy.

Further, we observe the fragment anion with mass of 110 u, which we assign to (FU–HF)^−^ anion with HF as a neutral counterpart. Four resonances are observed with maxima at 0.2 eV, 0.6 eV, and 1.7 eV and at 2.1 eV near the tail. The formation of a fragment anion with mass of 110 u has been also reported in uracil (abstraction of H_2_) [36] and 5-chlorouracil (abstraction of HCl) [34] upon electron attachment in the gas phase. Comparing our result to those reported in Ref. [4], the measured experimental resonance of 0.2 eV is in fair agreement. Considering the thermal energy stored in the molecule, the calculated threshold is in good agreement with the experimental one.

We suggest that HF formation commences with pre-dissociation of either H or F^−^ (Figure 3). After H crosses a pre-dissociation barrier, a small additional energy is required for dissociation. This energy is not captured correctly at the CCSD(T) level for the B3LYP-optimized structures; at the B3LYP/aug-cc-pVDZ level, the dissociation energy of a hydrogen atom from the pre-dissociated state is calculated as 0.09 eV (Figure 3). If the hydrogen atom does not dissociate directly, it might roam in the vicinity of the (FU–H)^−^ core and eventually recombine with F to form HF or take a position on the C5 carbon atom next to the F atom, creating the most stable structure with the intact ring at the energy of −0.89 eV.

The F^−^ anion might pre-dissociate from the molecule over a barrier of 0.72 eV compared to the energy of neutral FU (left-hand side of Figure 3); its full loss is, however, improbable at low electron energies as it requires, in total, 1.85 eV. However, it might go through a transition state with an additional barrier of 0.02 eV to form HF through reacting with a hydrogen atom of an N1–H bond. This reaction is exothermic, with an energy gain of −0.80 eV with respect to FU. The first part of this reaction pathway was analyzed previously [37]. From the final structure, HF might dissociate at the total energy of 0.12 eV, representing a low-lying dissociation channel. Note that after HF formation, the H loss channel is high in energy, and the presence of HF thus suppresses the H dissociation.

Our calculations enable us to explain the relative intensity of H and HF loss observed in the experiment. The HF molecule will be most easily formed at lower energies that already favor H pre-dissociation but do not allow for its full dissociation. Then, the system has enough time to head for the HF formation, possibly competing with the auto detachment of the electron. At higher energies, direct H loss prevails as a kinetically favored channel. If, on the other hand, F^−^ pre-dissociates, HF is most probably formed. Note that in the experiment, the H loss is preferred compared to HF loss by a factor of more than 100 already at the electron energy of ~0.7 eV.

We also observed electron-induced opening of the uracil ring via multiple bond cleavages, with lower intensities compared to the ion with mass of 129 u, as considerable molecular reorganization is needed. Suggested reaction pathways are depicted in Figure 3 and Figure 4. The heaviest fragment anion requiring ring dissociation is observed with mass 86 u in Figure 2c and could be assigned to HC_3_FNO^−^, NC_3_O^−^.HF, or H_2_C_2_N_2_O_2_^−^. For the first two fragment anions, an HCONH molecule is released and might further rearrange to NH_2_CO, gaining 0.77 eV of energy; if the H_2_C_2_N_2_O_2_^−^ anion is formed, HC_2_F might dissociate as a neutral counterpart:

C_4_FH_3_N_2_O_2_ + e^−^ → C_4_FH_3_N_2_O_2_*^−^ → HC_3_FNO^−^ + HCONH
(1)


C_4_FH_3_N_2_O_2_ + e^−^ → C_4_FH_3_N_2_O_2_*^−^ → NC_3_O^−^.HF + HCONH
(2)


C_4_FH_3_N_2_O_2_ + e^−^ → C_4_FH_3_N_2_O_2_*^−^ → H_2_C_2_N_2_O_2_^−^ + HC_2_F
(3)


The theoretical thresholds for reactions (1)–(3) are 1.99, 1.36, and 3.07 eV, respectively, while the experimental threshold is 3.2 eV. Thus, from the thermochemical perspective, the HC_3_FNO^−^/NC_3_O^−^.HF anions represent the more probable dissociation products. In a recent DEA study with uracil-5-yl O-sulfamate, Spisz et al. [38] reported the dissociation within the uracil moiety leading to the formation of H_2_C_2_N_2_O_2_^−^ which showed resonances within the energy range of 0–2 eV. However, here we observed resonances at higher energies, as summarized in Table 1, with the most intense peak recorded close to 6 eV. Formation of NC_3_O^−^.HF is also supported by the observation of the anion with mass of 66 u (Figure 5a) that has a relatively high intensity and is attributed to the loss of an HF molecule from the 86 u anion, forming NC_3_O^−^, see Figure 4.

The fragment ion at 82 u (Figure 2d) is attributed to a change from a six-membered to a five-membered ring over a transition state at 1.06 eV (Figure 4). In the formed anion with a five-membered ring, a CO group is attached to the ring and hydrogen-bonded by an HF molecule. If both CO and HF detach, we obtain the H_2_C_3_N_2_O^−^ ion with the reaction energy of 0.89 eV, below the experimental threshold of 1.3 eV.

The fragment anion with mass 62 u (Figure 5b) is most probably formed through direct dissociation of the ring after HF formation, leading to NCO^−^.HF, as shown in Figure 4. This assignment is supported by the observation of an anion with 42 u (Figure 6a), corresponding to further HF loss and formation of NCO^−^. The formation of NCO^−^ was reported by Abdoul–Carime et al. [5], which showed characteristic resonance shape and peak positions as those summarized in Table 1. Further support for the assignment is the fact that NCO^−^.HF and NCO^−^ ions exhibit a characteristic resonance around 7 eV, hinting that they are formed via the decay of the same TNI state. Previous studies [39,40] on electron attachment to the nucleobases, such as uracil and thymine, have shown that the NCO^−^ anion occurs as the second most abundant anion that follows the dissociation pathway of the dehydrogenated parent anion.

Fragment anions with masses 59 u, 58 u, and 39 u (Figure 5c,d and Figure 6b) are assigned to HC_2_N^−^.HF, C_2_N^−^.HF, and HC_2_N^−^, their anion efficiency curves show resonances in the broad energy region of about 4–10 eV. The three anions are again closely linked to HF formation as there is either HF present in the anion, or the anion is considered to be formed after HF loss. The anion with 59 u can be formed most easily (Figure 4), in agreement with its higher intensity in the DEA spectra, Figure 5c. The calculated thermochemical thresholds agree well with experimental appearance energies. The weakly abundant, broad feature at low electron energies in the ion yield for mass 59 u can be assigned to the artifact described above.

As mentioned earlier, the fragment anion with mass 26 u attributed to CN^−^ was already reported in Ref. [5]. The anion efficiency curve shown in Figure 6c exhibits similar resonance positions as reported in Ref. [5], except for an additional resonance which we observe around 1.8 eV. Although the appearance of the anion in this energy range seems to be possible given the calculated threshold of 0.77 eV, the peak might be again artificial.

Finally, a major interest of this study was to investigate the formation of the halogen anion F^−^. The anion efficiency curve of F^−^ is shown in Figure 6d. From the shape of the curve, we observe an anion yield below 3.6 eV, which we attribute to an artifact (see below). Above this energy, the anion yield curve exhibits a broad feature between 4 eV to 14 eV with several overlapping resonances. We observe an experimental threshold of 3.6 eV for the main feature of F^−^ anion yield (Table 1). Our quantum chemical calculations offer three pathways for F^−^ loss (see Figure 3): (i) a direct dissociation of the C–F bond in the FU valence anion with the energy of 1.85 eV compared to neutral 5-FU, in agreement with previous computational studies [32,41,42]; (ii) a pathway starting with H pre-dissociation and its attachment to the carbon of the C–F bond in a very stable conformation at −0.89 eV (the carbon atom of the dissociating C–F bond is then saturated by the hydrogen atom, dropping the energy required for F^−^ dissociation to 1.02 eV); (iii) F^−^ release coupled to the formation of a shape resonance. As indicated by previous works, nucleobases form low-energy shape resonances (for instance, the work from Illenberger’s and Märk’s groups [43] shows a sequence of shape resonances leading to the cleavage of the N–H bond in thymine and its deuterated forms). On the other hand, resonance strand break (SB) formation in DNA [9] occurs at much higher energies of an incident electron (around 10 eV) than the energies of π* orbitals in nucleobases (determined with electron transmission spectroscopy [9]). This discrepancy suggests, thus, that electrons whose attachment to DNA results in SB form core-excited resonances rather than shape resonances. In order to interpret the low energy F^−^ yield shown in Figure 6d, we used a variant of the extrapolation method that enables dealing with metastable anions using conventional electronic structure tools. Namely, we introduced an additional artificial charge (Δq) on the C5 nucleus in FU to make the studied states electronically stable. Then, we calculated the vertical electron affinity (VEA) for a set of charges equal to or larger than that which makes the studied anion electronically stable and, finally, extrapolated the obtained dependency, VEA = f(Δq), to VEA at Δq = 0, obtaining thus an estimate of the resonance energy E_R_ of the studied metastable state. In this way, we calculated two shape resonances (Figure 7a,b) that explain the experimental 4.4 and 7.4 eV F^−^ yields (Table 1 and Figure 6d). The shapes of the singly occupied molecular orbital (SOMO), see Figure 7, suggest that the 4.4 eV resonance (calculated as 4.4 eV as well) is related to electron attachment to the unoccupied N1–H σ* orbital [44] coupled with the σ* C–F orbital (Figure 7a). A similar indirect mechanism of the dissociation process was postulated before to explain DEA to nitroimidazole [45]. The second resonance is, in turn, related to the direct attachment of the excess electron to the σ* C-F orbital (Figure 7b). In order to describe higher lying resonances (above 10 eV, see Figure 6d), one needs a different methodology that could characterize core-excited resonances and/or involvement of the Rydberg-type states. Previously, a core excited Feshbach resonance was identified at about 6 eV for uracil [46].

We suggest that dissociation of the F^−^ anion is substantially lowered by its high reactivity as it might easily form HF with an H atom instead of dissociating. This explains the low F^−^ signal compared to the overall signal intensity of anions formed after HF formation. We further note that of all anions observed, the F^−^ yield most clearly shows the artificial ion yield raising below the predicted thermochemical threshold, see Figure 6d. This signal remains almost constant till the main resonant features appear above 3.6 eV. The relatively strong abundance of this artificial signal may arise due to the extended energy range of F^−^ formation, i.e., the energy window for attaching an electron formerly released from the parent anion is comparably large in this case.

## 3. Methods and Materials

The present experiment is performed in a crossed electron-molecular beam setup described in detail in Refs. [47,48]. A monochromatized electron beam generated by an electrostatic hemispherical electron monochromator interacts perpendicularly with an effusive beam of fluorouracil molecules. The 5-FU beam is generated by heating the 5-FU powder sample in an oven up to 134 °C and effusing the sublimated molecules through a 1 mm capillary directly into the collision region. The anions resulting from the electron-molecule collisions are extracted from the collision region by a weak extraction potential and focused onto a quadrupole mass filter where they are mass analyzed. The anions are detected by a channeltron secondary electron multiplier combined with single-pulse counting electronics. To record the shown anion efficiency curves (the raw data can be found in the Appendix A), the mass was kept constant and the electron energy was varied The electron energy scale and energy resolution have been determined by measuring the zero-eV peak in the Cl^−^ anion yield upon electron attachment to CCl_4_. The current of electrons was about 10–70 nA. The energy resolution was determined to be 100 meV at full width at half maximum (FWHM). The sample with a purity of 98% was purchased from Sigma Aldrich, Austria, and was used as delivered. The base pressure of the interaction chamber was in the range of 10^−8^ mbar. The working pressure was about 2.1 × 10^−7^ mbar.

Molecules and ions along 5-FU dissociation pathways were calculated using density functional theory (DFT) at the B3LYP/aug-cc-pVDZ level [49,50,51,52], with wave function stabilization performed prior to each optimization. To obtain more reliable energetics, the resulting structures were single-point recalculated using coupled cluster singles and doubles with non-iteratively included triples, CCSD(T)/aug-cc-pVDZ [53]. Reported reaction energies include zero-point energies calculated at the B3LYP/aug-cc-pVDZ level. To describe a dipole-bound state, we used the aug-cc-pVDZ basis set on heavy atoms and aug-cc-pVTZ on hydrogens, with two additional *s* functions and one additional *p* and *d* function for hydrogens, with coefficients determined as one-third of the lowest coefficient of the respective *s*, *p*, and *d* functions in the basis set. This basis set is denoted as aug-cc-pVDZ(C,N,O,F),TZ(H)+. To estimate the position of metastable resonances, we applied the extrapolation method [54], which was described in detail in our previous work [55]. In short, we carried out the optimization of the neutral 5-FU geometry at the MP2 level [56] with the cc-pVDZ basis set [51]. Then, the nuclear charge of the C5 atom was increased by Δq values in order to make the studied system electronically bound. This way, we obtained a series of bound electronic states in which the SOMO possesses C5-F antibonding character. Next, the electron binding energy, D, i.e., the difference between the energy of the neutral and anion radical, was plotted against additional Δq charge at C5. Finally, in order to estimate E_R_, we extrapolated D to Δq → 0. For evaluation of cross sections, more advanced approaches would be needed, e.g., the *R*-matrix method [57,58]. All calculations were performed with the Gaussian quantum chemical package [59]. Cartesian coordinates and energies of optimized isomers can be found in the Appendix A.

## 4. Conclusions

In the present study, we have carried out a detailed investigation of (dissociative) electron attachment to 5-fluorouracil. Experimentally, we found eight fragment anions not reported in the literature so far. A detailed computational study provided novel insight into reaction pathways. The present results suggest a strong involvement of hydrogen fluoride formation within the dissociation processes upon DEA to 5-fluorouracil, with the HF molecule appearing in many dissociation pathways. HF formation is initiated by pre-dissociation of an H atom that might either leave the molecule directly or roam in its vicinity, as suggested previously for an OH group in tirapazamine [60,61], and finally recombines to HF. Formation of the halogen anion F^−^ is a minor but clearly observable channel in the experiment. Our calculations show that F^−^ might indeed be formed above electron energies of ~1 eV, though we assign the ion yield found at this energy to an artifact. The weak F^−^ yield observed between 4 and 10 eV is due to shape resonances, while the signal above 10 eV might be related to core-excited species. The fluorine anion, however, also efficiently forms HF before it can leave the molecule.

We could only find hints on the metastable parent anion FU^−^ in our experiment, while for other halouracils, the parent anion in the gas phase seems longer-lived since it is directly observable by mass spectrometry [34]. Interestingly, the reported lifetime of the FU^−^ anion in solution (>15 µs) turned out to be much longer compared to other halouracils studied [62]. Moreover, solution phase studies indicated that FU^−^ dissociated less readily upon the interaction with hydrated electrons than the other halouracils [62,63]. In this case, it was suggested that stabilization of the FU^−^ anion by protonation efficiently occurs upon capture of a hydrated electron [62]. In contrast, under isolated conditions in the gas phase, spontaneous electron emission seems to be prevalent upon capture of an electron with thermal energies, which can be explained by the electron affinity close to zero eV and the endothermicity of nearly all dissociation channels.

Although at first glance, the current studies are only weakly bound to the radiosensitizing properties of substituted uracils, one has to admit that the gas-phase results allow one to deeply understand the process that lies behind the radiosensitizing properties of substituted pyrimidines. Indeed, according to the generally assumed radiosensitization mechanism [17], a substituted uracil, after being incorporated into the cellular DNA, has to attach solvated electron that induces DEA in nucleobase, ultimately leading to a reactive uracil-5-yl radical. Of course, solvent modifies the dissociative behavior of the formed uracil anion, but if the anion did not undergo DEA in the gas phase, it would probably be an inefficient radiosensitizer. So the gas-phase studies deliver the very first evidence of radiosensitizing properties. In this sense, the results of gas-phase measurements are relevant for radiolysis taking place in liquid environments. Here one could also formulate a general conclusion that results from our previous studies concerning a number of uracil derivatives. For substituted uracil to be a potential radiosensitizer, the C5 substituent should be an electron-withdrawing group in order to facilitate trapping of the excess electron by a nucleobase while the chemical bond between the substituent and C5 (or bond(s) within the substituent) should be sufficiently weak to allow for an efficient DEA process, i.e., for the formation of reactive species within DNA, leading to its damage [64].

## Figures and Tables

**Figure 1 ijms-23-08325-f001:**
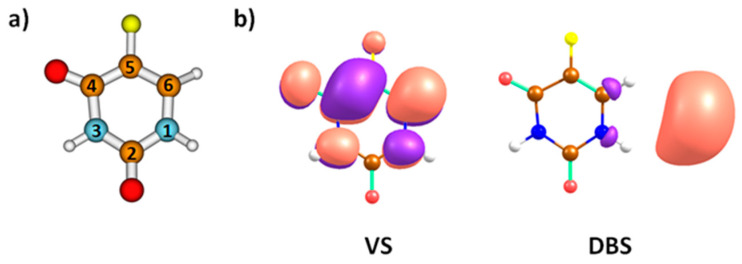
(**a**) Local minimum structure of 5-FU as optimized at the B3LYP/aug-cc-pVDZ level. (**b**) Singly-occupied Hartree-Fock orbitals in the lowest-lying valence state (VS) and the dipole bound state (DBS) within CCSD(T)/aug-cc-pVDZ and CCSD(T)/aug-cc-pVDZ(C,N,O,F)TZ(H)+ calculations, respectively. Wavefunction phases are shown in purple and salmon-pink. Color code: carbon—brown; nitrogen—blue; fluorine—yellow; oxygen—red; hydrogen—grey.

**Figure 2 ijms-23-08325-f002:**
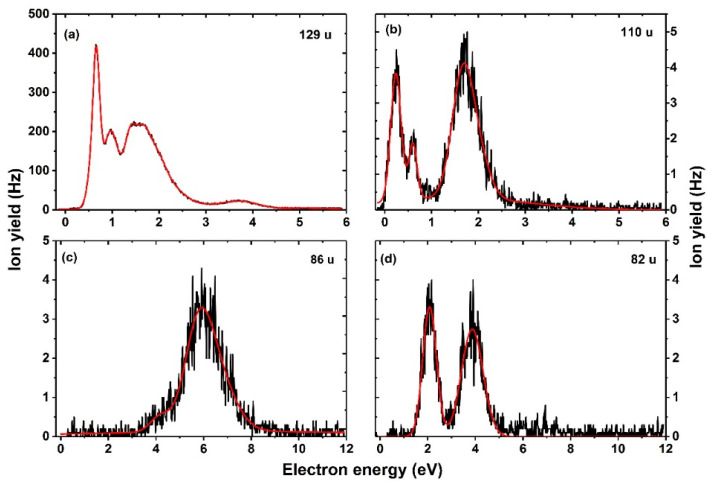
Anion efficiency curves of fragments formed with masses (**a**) 129 u, (**b**) 110 u, (**c**) 86 u, and (**d**) 82 u upon electron attachment to 5-FU. The red line represents the cumulative peak from the Gaussian fit.

**Figure 3 ijms-23-08325-f003:**
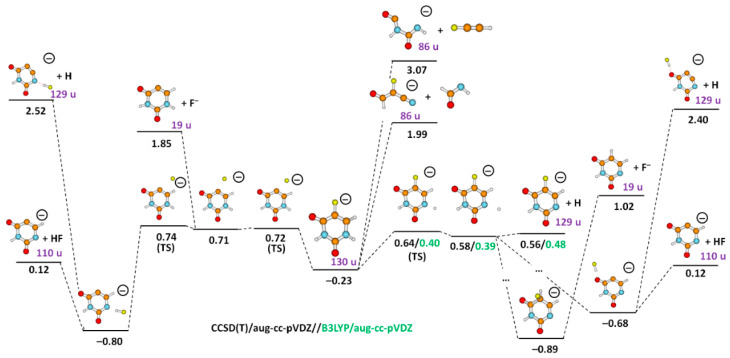
Simplified reaction pathways after electron attachment to FU, forming FU^−^ (the central anion at −0.23 eV). Ion mass is included for FU^−^ and experimentally observed ions. The energy is given in eV with respect to the neutral FU molecule as calculated at the CCSD(T)/aug-cc-pVDZ//B3LYP/aug-cc-pVDZ level. For the H pre-dissociation pathway, small numerical discrepancies between energies of local minima and transition states or dissociation limits are induced through CCSD(T) recalculation of B3LYP-optimized structures; therefore, both CCSD(T) and B3LYP values are given. Color code: carbon—brown; nitrogen—blue; fluorine—yellow; oxygen—red; hydrogen—grey.

**Figure 4 ijms-23-08325-f004:**
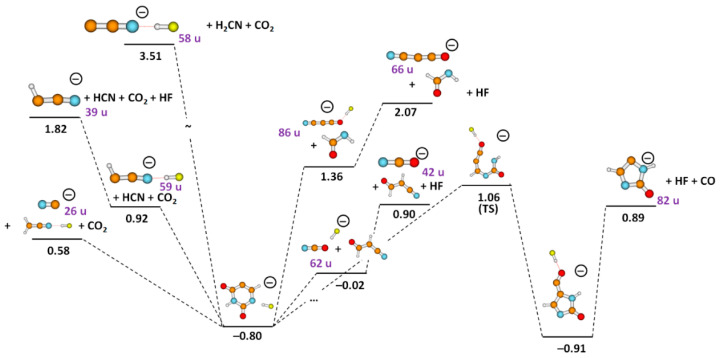
Simplified reaction pathways after formation of HF molecule within FU^−^. Ion mass is included for experimentally observed ions. The energy is given in eV with respect to the neutral FU molecule as calculated at the CCSD(T)/aug-cc-pVDZ//B3LYP/aug-cc-pVDZ level. Color code: carbon—brown; nitrogen—blue; fluorine—yellow; oxygen—red; hydrogen—grey.

**Figure 5 ijms-23-08325-f005:**
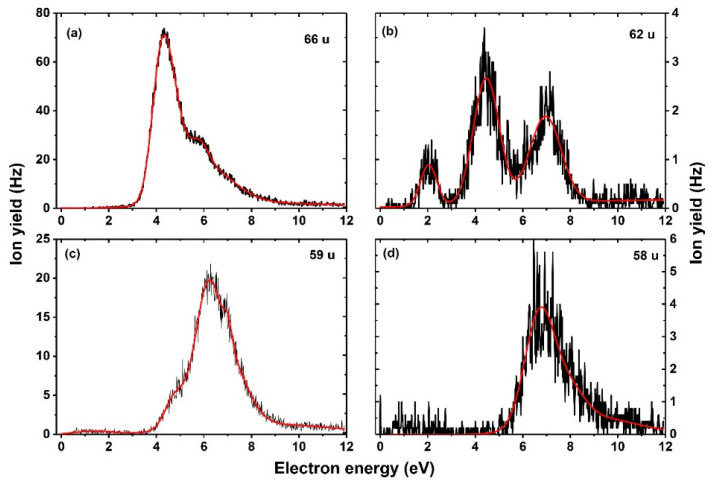
Anion efficiency curves of fragments formed with masses (**a**) 66 u, (**b**) 62 u, (**c**) 59 u, and (**d**) 58 u upon electron attachment to 5-FU. The red line represents the cumulative peak from the Gaussian fit.

**Figure 6 ijms-23-08325-f006:**
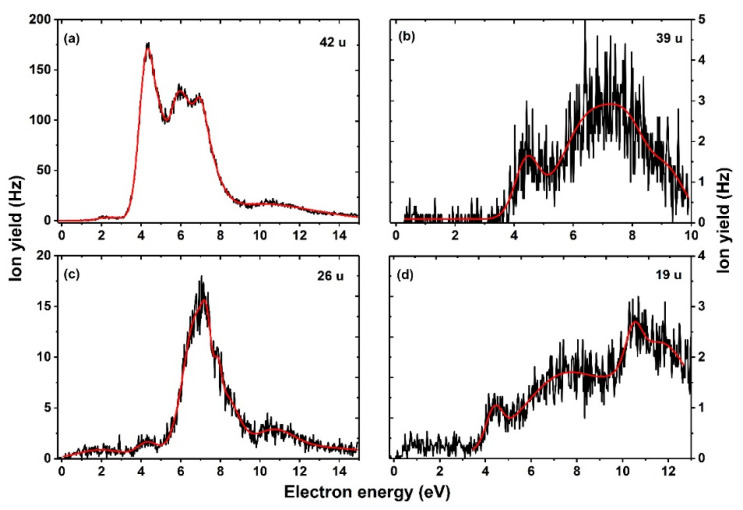
Anion efficiency curves of fragments formed with masses (**a**) 42 u, (**b**) 39 u, (**c**) 26 u, and (**d**) 19 u upon electron attachment to 5-FU. The red line represents the cumulative peak from the Gaussian fit.

**Figure 7 ijms-23-08325-f007:**
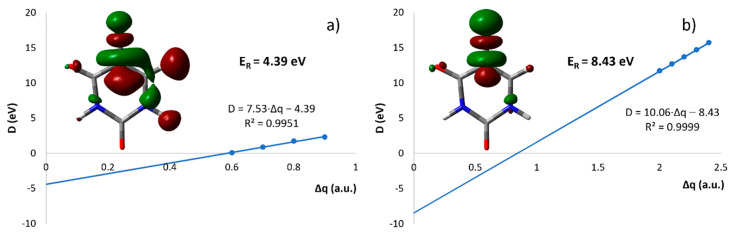
Binding energy, D, plotted against the stabilizing charge, Δq, values for which the anion is electronically stable and extrapolated to Δq → 0. SOMO orbital visualized with the contour value of 0.05 (a_0_)^−f3/2^ for Δq = 0.7 (**a**) and for Δq = 2.4 (**b**). E_R_ stands for the calculated position of the resonance related to the release of the fluoride anion.

**Table 1 ijms-23-08325-t001:** Summary of fragment anions in terms of their masses, structures, resonance positions, and experimental and calculated thresholds upon electron attachment to 5-FU. Calculations were performed at the CCSD(T)/aug-cc-pVDZ//B3LYP/aug-cc-pVDZ level of theory. Three threshold energies for the ion with mass of 86 u refer to three different ion products (see text).

Mass (u)	Anion	Resonance Maxima (eV)							Threshold (eV)	
		1.	2.	3.	4.	5.	6.	7.	Exp.	Theory
129	C_4_FH_2_N_2_O_2_^−^	0.6	0.7	1.0	1.4	1.6	2.1	3.7	0.4	0.56
110	C_4_H_2_N_2_O_2_^−^	0.2	0.6	1.7	2.1	–	–	–	~0	0.12
86	HC_3_FNO^−^/NC_3_O.HF^−^/H_2_C_2_N_2_O_2_^−^	4.1	5.9	7.2	–	–	–	–	3.2	1.99/1.36/3.07
82	H_2_C_3_N_2_O^−^	2.1	3.9	–	–	–	–	–	1.3	0.89
66	NC_3_O^−^	4.2	4.7	5.8	6.1	8.7	–	–	3.2	2.07
62	NCO^−^.HF	2.0	4.5	7.1	–	–	–	–	1.2	−0.02
59	HC_2_N^−^.HF	4.8	6.1	7.0	8.7	–	–	–	3.5	0.92
58	C_2_N^−^.HF	6.8	7.2	7.9	8.3	–	–	–	5.2	3.51
42	NCO^−^	2.3	4.3	5.8	7.0	7.9	10.3	–	0.9	0.90
39	HC_2_N^−^	4.4	6.6	7.9	9.2	–	–	–	3.4	1.82
26	CN^−^	4.3	6.8	7.2	7.9	8.3	10.7	–	3.2	0.58
19	F^−^	4.4	7.4	10.6	11.8	–	–	–	3.6	1.02

## Data Availability

The raw data are included in the Appendix A of this article.

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
