# Peer review of "Electron Attachment to 5-Fluorouracil: The Role of Hydrogen Fluoride in Dissociation Chemistry"

_ijms, 2022, doi:10.3390/ijms23158325_

Round 1

Reviewer 1 Report

Referee report

Title of the manuscript: Electron Attachment to 5-Fluorouracil: The Role of Hydrogen Fluoride in Dissociation Chemistry

Authors: Eugene Arthur-Baidoo, Gabriel Schöpfer, Milan Ončák, Lidia Chomicz-Mańka, Janusz Rak and Stephan Denifl

Manuscript ID: ijms-1820302

In the submitted manuscript the Authors report results on dissociative electron attachment process for Fluorouracil molecule. In particular, they investigate the process from theoretical as well as from experimental point of view. The Authors observe 12 different final states, including 8 dissociative channels never reported before in literature. For each of them, they analyze with accuracy the principal paths of dissociation. The main result of the paper relay on a strong formation of hydrogen fluoride in the dissociation processes for Fluorouracil, with the HF molecule appearing in many dissociation pathways.

The manuscript, in general, is well written, the theoretical model and the experimental set up clearly presented and the results correctly analyzed. I think that the paper itself is up to the standard of the journal and it is well aligned to its scientific scope.

Concerning theoretical model of electron capture following by dissociation process, I have some questions to address to the authors:

-       Concerning the measured “Ion yield” plots on figures 2, 5 and 6 they are linked for sure to the cross sections of the dissociative electron attachment process. Are the authors able, within the presented theoretical model, to estimate the cross sections for e + FU dissociation?

-       For the energetic diagram in figures 3 and 4, I guess were calculated at equilibrium geometry for FU molecule. Is it possible in some ways to take into account molecular vibration and/or rotation in the presented theoretical model?

-       Normally, in order to calculate dissociative electron attachment or more generally the resonant scattering of molecules by electron impact, is need to include in the basis sets the continuum orbitals which describe correctly the incoming electron as in the Rmatrix approach. See for example the papers which could be cited:

V. Laporta, I.F. Schneider and J. Tennyson, “Dissociative electron attachment cross sections for ro-vibrationally excited NO molecule and N- anion formation”, Plasma Sources Sci. Technol. 29, 10LT01 (2020)

K. Chakrabarti, V. Laporta and J. Tennyson, “Calculated cross sections for low energy electron collision with OH”, Plasma Sources Sci. Technol. 28, 085013 (2019)

            can the Authors explain how the incident electron is included in the theoretical model and how the resonances calculated?

I think the manuscript can be accepted after the Authors reply to these questions.

Author Response

See the word file attached

Reviewer 2 Report

From the perspective of a reader who's expertise lies outside the subject of this work, generally I find this to be a well written manuscript.  A study of dissociative fragmentation of 5-fluorouracil resulting from electron attachment is described in detail.  Measured efficiency curves as a function of electron energy are shown for various fragment anions.  These, combined with theoretical information, are used to predict/analyze the various reaction pathways.  Although the interpretation of these pathways is dependent on various theoretical values which I do not have the expertise to evaluate, the methods and interpretations are well explained and appear to be valid.  However, there are two areas that I found lacking in reading the manuscript.  One has to do with radiation sensitivity which is the primary topic of the introduction.  It is stated (the following is highly paraphrased) that cleavage of a C5-X bond leads to formation of radicals which play a role in single strand breaking of DNA.  Based on this, for me major questions would be "what do the present results have to do with this?; are the present gas-phase results relevant to liquid-phase environments? what happens if X is changed?; can the results of the present study be used to either understand, or improve, radiation sensitivity of similar compounds?"  Some of these questions are particularly addressed in the conclusions but no suggestions as to directions to pursue in future studies or which X may be better and why are provided, as I would expect from reading the introduction.  The other area concerns the relative probabilities of the various pathways.  Evan after a scanned reading following a complete reading of the manuscript, I find little or nothing being said about the relative probabilities, i.e., the ordinate axis in figures 2,5,6.  Rather, most of the discussion is about how different anions are formed with little or no discussion as to which pathways are most probable, i.e., information from Figures 2, 5, 6. In my opinion, the relative importance is just as important as determining the reaction pathways.  Along with this, as a suggestion, it would be very useful to somehow (e.g. line weight or color) designate the reaction probability in Figures 3 and 4.  My recommendation is that the authors consider these comments and submit a revised manuscript for further review.  

Specific comments:

Abstract:  "on the experimental timescale"    It would be helpful to some readers to give a numerical value. 

Table I:  In many cases the discrepancies between the experimentally measured and theoretically predicted thresholds is considerable.  The discrepancies are larger than what I would expect could be attributed to experimental uncertainties.  Adding some comments about this seems appropriate.  Also, for mass 86, three theoretical values are given.  I assume these are from the different level of theory referred to in the caption.   Reading further, this may be partially addressed on page 5.  However, this should also be made clear as the reader looks at Table I.  

Figure 1:  In part b I am not sure what the purple and ?rose brown? are.  Purple isn't listed in the caption and the ?rose brown? in part b is not the same color as the brown carbon atoms in part a.  From the text and caption, I assume they are atoms.  But, from the structure, they could be electron clouds.  

Figures 3 and 4:  Please add the color scale here so the reader does not have to go back to Figure 1 to interpret the figure.

Author Response

See the word file attached

Round 2

Reviewer 1 Report

The Authors replied to the comments and the new attached version improuve the quality of the manuscript. I suggest for the acceptance of the paper

Reviewer 2 Report

I thank the authors for addressing my previous comments.  In particular, I think, and I hope the authors agree, that the added paragraph in the conclusion greatly improves the significance of this study.  I recommend acceptance of the present version for publication without further review.